# Methods of Formation of Protective Inhibited Polymer Films on Tungsten

**DOI:** 10.3390/ijms241914412

**Published:** 2023-09-22

**Authors:** Natalia A. Shapagina, Alexey V. Shapagin, Vladimir V. Dushik, Andrey A. Shaporenkov, Uliana V. Nikulova, Valentina Yu. Stepanenko, Vladimir V. Matveev, Alexey L. Klyuev, Boris A. Loginov

**Affiliations:** 1Frumkin Institute of Physical Chemistry and Electrochemistry, Russian Academy of Sciences, Leninsky Prospect 31-4, 119071 Moscow, Russia; shapagin@mail.ru (A.V.S.); v.dushik@gmail.com (V.V.D.); shipr24@mail.ru (A.A.S.); ulianan@rambler.ru (U.V.N.); 4niko7@list.ru (V.Y.S.); matveev46@yandex.ru (V.V.M.); klyuevchem@mail.ru (A.L.K.); 2Research Laboratory of Atomic Modification and Analysis of Semiconductor Surface, National Research University of Electronic Technology (MIET), Shokin Square, Bld. 1., 124498 Zelenograd, Russia; b_loginov@mail.ru

**Keywords:** tungsten, inhibited polymer films, cataphoretic deposition (CPD), organosilanes, corrosion inhibitors, inhibited formulations (INFOR)

## Abstract

A comparative study of anticorrosive inhibited polymer films on the tungsten surface formed from an aqueous solution of inhibited formulations (INFOR) containing organosilane and corrosion inhibitors was carried out by means of the prolonged exposure of a tungsten product in a modifying solution and by the method of cataphoretic deposition (CPD). Depending on the method of forming films on tungsten, the molecular organization of the near-surface layers was studied (ATR-FTIR), and the subprimary structure of the films was explored (TEM). The optimal modes of cataphoresis deposition (CPD duration and current density applied to the sample) for the formation of a protective inhibited polymer film on the tungsten surface were established by means of SEM. The energy and thermochemical characteristics (sessile drop and DSC methods), as well as operational (adhesive behavior) and protective filming ability (EIS and corrosion behavior), according to the method of formation of inhibited polymer film, were determined. Based on the combined characteristics of the films obtained by the two methods and the deposition modes, the CPD method showed better performance than the electroless dipping method.

## 1. Introduction

Tungsten and tungsten-based coatings are very promising wear-resistant and anticorrosive materials that can be used to protect important components of aerospace, chemical, oil and gas, and other industries [1,2,3,4]. Upon contact with an aggressive medium (pH 0–5), a protective film of WO_3_ or W_2_O_5_ oxides is formed on the metal, which improves its corrosion resistance in such environments [1,3]. However, in neutral and alkaline solutions and humid environments (pH > 6), the corrosion resistance of tungsten decreases, especially in oxidizing or aerated media. Under these conditions, tungsten ionization occurs in the form of an acidic residue of tungstic acid, which leads to active dissolution of the metal [1,2,4]. Thus, tungsten-based materials need additional protection in pH–neutral atmospheric conditions (pH 5.5–7.5). Chromate coatings could be used as an additional and temporary protection of tungsten-based materials; however, their use is excluded due to toxicity [5,6,7,8,9,10,11,12]. Since there is no information about corrosion inhibitors for the protection of tungsten products in the modern literature, studies on the inhibition of tungsten corrosion remain topical. In this regard, one of the options for protecting the tungsten surface is the use of inhibited formulations (INFOR) consisting of molecules of organosilane and corrosion inhibitors. Organosilane is the main film-forming component because of its ability to polycondensation. Corrosion inhibitors are added to the solution to inhibit the corrosive process. In the result, from aqueous compositions INFOR on the metal surface are formed inhibited polymer films. In these films, the organosilane layer is a barrier layer protecting the metal from the influence of aggressive environments. If the film solidity is disturbed, the corrosion inhibitors inhibit the corrosion dissolution of the metal [13,14,15]. 

Thus, in [13], siloxano-azole fragments were shown to be formed on the metal surface using an aqueous chloride-containing solution of a vinyltrimethoxysilane or diaminsilane and 1,2,3-benzotriazole composition. This provides additional crosslinking of surface-adsorbed molecules, thereby increasing the density of the chemical mesh and, therefore, the density of the surface layers. Also, films on a metal surface have the possibility of being formed using INFOR to protect metals (steel and copper) in various corrosive media [14,15]. The presence of carboxylic or phosphonic acids in the modifying solution leads to additional interaction with siloxane groups, which makes for a thicker film, and as a result, higher corrosion resistance [15].

The traditional method of the production of inhibited polymer films on a metal product is the method of prolonged exposure of the metal sample in a modifying solution of INFOR [16,17,18]. The practical experience proves that this approach has disadvantages. The major problem is that the protective effect of the film is defined by holding time of the material in the INFOR solution (coating solidity, uniformity, adhesion properties, etc.) [15]. An alternative method of obtaining a higher-quality coating/film is electrodeposition—application of the coating/film under the action of a constant electric field. As a result, charged particles migrate and surface sediment on an oppositely charged electrode [19]. Electrophoretic deposition (EPD) is subdivided into cataphoretic and anaphoretic deposition [19,20]. Coatings formed on a metal surface using electrodeposition have a number of advantages compared to the immersion/exposure method: these coatings can be applied from aqueous polymer dispersions to metal substrates of any configuration; the coatings have high adhesive strength, and the formed coatings are denser, uniform, and less defective [20,21].

Typically, the EPD method is used for materials already pre-treated with organosilane or added to sol–gel coating solutions. In the case of pre-silanization, organosilane is used as an intermediate layer, between the metal substrate and the main coating, to improve the adhesion properties of the main coating [22,23,24,25]. In the case of sol–gel coatings, the solutions usually contain one or more organometallic compounds, such as zirconium, aluminum, or titanium compounds, one or more organosilanes, and acids, bases, glycols, etc., are used as catalysts [26,27,28]. The addition of organosilanes produces more dense particles, and the sols themselves become more viscous, which, as a result of further electrodeposition, allows for the obtaining of virtually defect-free coatings.

Thus, the research objective was to compare the structural feature, protective ability, and stress–strain behavior of the films formed prolonged exposure of a tungsten product in a modifying solution and cataphoresis deposition on the surface of tungsten from the INFOR aqueous solution.

## 2. Results

### 2.1. Determination of Optimal Modes of Film Formation by CPD

Using scanning electron microscopy, we were able to define the optimization conditions of the CPD (Figure 1 and Figure 2).

Figure 1 (1 and 2.5 min of CPD) shows the occurrence of a film on the tungsten surface, but it was not continuous and had an “insular” appearance. The average thickness of the film was 3–5 µm. Increasing the duration of cataphoresis to 10 and 20 min led to the production of a solid film with a thickness of 19–24 µm. However, as can be seen from Figure 1 (10 and 20 min of CPD), the coating had defects (the shrinkage cracks, and in some places, the film detached from the tungsten base). It was found experimentally that in 5 min of CPD, the film formed on the sample was continuous and defect-free, and its thickness was 8–10 µm.

Since 5 min of cataphoresis deposition is the optimal time to form a continuous and dense film, we compared the morphological properties of the film when the sample was exposed in the INFOR aqueous solution for 5 min. As shown in Figure 1 (1 and 2.5 min of CPD), an “insular” film was formed on the sample, and its thickness was 1–2.3 µm. The prolonged exposure of tungsten product up to 60 min in the modifying solution resulted in the creation of a film on the metal similar in morphology to the film obtained in 5 min of CPD.

It was found experimentally that holding the sample in the INFOR aqueous solution for 5 min resulted in the appearance of the “insular” film on the metal (Figure 2) (currentless). The solid film was formed on the sample at current densities 0.5 ÷ 1.5 A/dm^2^. At current densities of more than 1.5 A/dm^2^, shrinkage cracks appeared in the film.

### 2.2. Film Structure

#### 2.2.1. The Molecular Behavior

According to Section 2.5, the molecular behavior of the ingredients of the inhibited polymer films made by different methods was studied (Figure 3).

The FTIR spectrum for the studied MS corresponded to those previously presented in the literature [29]. The asymmetric and symmetric vibrations of the C–H bonds were clearly visible from the FTIR spectrum (2945 and 2841 cm^−1^). In the spectrum, stretching vibrations were detected: 1715 cm^−1^ for the carbonyl group, 1638 cm^−1^ for the double bond of C=C, 1200–1100 cm^−1^ for the ester group C–O–C, and 899 cm^−1^ for the bond of C–C. The area of interest was 1000–814 cm^−1^, in which we detected the stretching and the deformation vibrations for the Si–O–C, Si–C, and Si–O–CH_3_ groups.

FTIR spectra of the films in general correlated well with the spectrum of the original organosilane but had a number of fundamental differences. First, in the region of 3400–3500 cm^−1^, the appearance of hydroxyl vibrations was clearly traced. It was maximally manifested in the spectrum corresponding to the film obtained when the sample was kept in the INFOR aqueous solution for 60 min (Figure 3, red spectrum). Second, a decrease in the peaks’ intensity in the 2800–3000 cm^−1^ range and a shift of the 2841 cm^−1^ band to the 2895 cm^−1^ position was observed. Third, there was a broadening of the stretching vibration band of the carbonyl group at 1715 cm^−1^. Finally, there were significant changes in the ν(C–C), ν(C–O–C), ν(Si– O–C), ν(Si–C), and δ(Si–O–CH_3_) vibration bands, presented in more detail in Figure 4. All the listed changes clearly indicate the hydrolysis of organosilane with the replacement of methyl groups by hydroxyl groups according to the first step of the scheme presented in Figure 3. This clearly explains the appearance of hydroxyl groups, the decrease in –CH_3_ groups, and the sharp drop in the intensity of Si–O–CH_3_ bond strain vibrations at 814 cm^−1^. The characteristic bands for BTA and HEDP, which were used in the formation of the investigated films, were absent in the spectra of the films. This was confirmed by a comparative analysis of the FTIR spectrum of the films with and without their presence. This can be interpreted by their small number. Figure 4 shows the region from 800 to 1200 cm^−1^ in more detail.

The redistribution was clearly visible, and the characteristic band intensities were a product of the above-described hydrolysis of MS. The transformation of Si–O–CH_3_ groups into Si–OH was accompanied by for the following: 3433 cm^−1^ for the appearance—OH groups, 1100–1050 cm^−1^ for replacement of ν(Si–O–C) by ν(Si–C), and intensity reduction for the band at 814 cm^−1^. Si–O–CH_3_ groups were transformed into Si–OH, which was manifested as a decrease in the band intensity and change in the band intensity at the hydroxylate appearance. The conformational structure of the radical in organosilane noticeably changed, because ν(C–C) bond (900–950 cm^−1^) and the C–O–C ester group (1100–1200 cm^−1^) changed. But special attention should be paid to the region of 1010–1030 cm^−1^, where the coatings showed the appearance of a broad double peak related to the appearance of the Si–O–Si complex group. This was due to the cross-linking of hydrolyzed silane on the tungsten, as shown in the scheme (Figure 3, stage 2th). In this case, the difference in different methods of coating was manifested most strongly in a marked decrease in the hydroxyl groups at CPD.

#### 2.2.2. Permolecular Structure of Films

The phase structure of the protective films was studied by transmission electron microscopy. In both coatings, a crystalline BTA phase with the size of structural elements from 50 to 100 nm was identified (Figure 5).

To confirm that the dispersed particles presented in Figure 5 corresponded to the BTA phase, we additionally obtained TEM images (Figure 6) of an aqueous solution of BTA with a concentration of 0.01 M. It was found that after drying at room temperature, similar dispersed particles were identified on the carbon substrate, just as in the study of organosilane systems modified with BTA, which confirmed the presence of BTA in INFOR compositions in the insoluble crystalline state.

Thus, it can be stated that when a film was formed from the INFOR aqueous solution by soaking the sample in the INFOR solution for one hour, the crystalline phase came to the surface (Figure 5) (60 min of exposure), whereas with the CPD, the crystallites were in the film volume (Figure 5) (5 min of CPD).

### 2.3. Phase Transitions

The change in phase transition temperatures during the formation of the protective film is shown in Figure 7.

The initial MS and studied systems after curing were characterized by the glass transition temperature. The endothermic melting peaks of BTA and HEDP were not identified due to their low concentrations in the solution and the small amount of the test object in the DSC crucible. Figure 7 shows that the glass transition temperature of the initial MS was −71 °C. The formation of films on the tungsten surface for both methods increased T_g_ (glass transition temperature) up to 70 °C, which confirmed the formation of mesh structures in the process of polycondensation. The film obtained by cataphoretic deposition was characterized by a slightly lower (by 3 degrees) glass transition temperature and correspondingly a rarer mesh-like permolecular structure, which was probably caused by steric hindrances in the form of crystal structures of BTA to the formation of a three-dimensional chemical bond network (Figure 6).

### 2.4. Energy Characteristics of the Obtained Films

The data of measuring the surface energy are submitted in Table 1.

It was established that the films were characterized as being sufficiently high in terms of polymers with the general surface energies of 37.2 and 42.23 mJ/m^2^ and had comparable values in terms of their polar components that were indicative of the similarity of the functional groups on the films’ surfaces. Some difference in the dispersive component was recorded, which indicated denser structures of the near-surface coating layers from the solution. Probably, this was caused by the appearance of crystalline structures on the surface and a denser network of chemical bonds.

### 2.5. Study of the Adhesive Properties of Films

The results of adhesion studies depending on the method of film formation are shown in Figure 8.

The adhesion diagrams show that at a given peel rate, the film/scotch system broke down due to the adhesion mechanism, i.e., without breaking the continuity of the protective film. Average peel resistance values up to 357 N/m (Figure 8a) and 333 N/m (Figure 8b) ensured the reliable operation of the films at these peel stresses. The higher peel resistance values of the tape from the INFOR solution film were explained by the higher surface energy values due to the more densely structured surface layers.

### 2.6. Evaluation of the Barrier Properties of Films Formed on Tungsten Depending on the Method of Film Formation

#### 2.6.1. Electrochemical Behavior of Films

The use of electrochemical impedance spectroscopy allowed for a deeper understanding of the processes occurring in the films. Figure 9a shows Bode diagrams for the typical impedance spectra of the studied systems.

According to the phase-frequency dependence, the equivalent circuit contains one relaxation time for all three types of electrodes. This relaxation time is the Voigt link, consisting of the polarization resistance R_p_ and the capacitance of the double electric layer C_dl_. In this case, because of the roughness of the electrode surface, the capacitance of the double electric layer is modeled by the constant phase element CPE_dl_, $Z_{CPE}\left(\omega \right)=\frac{1}{T{\left(j\omega \right)}^{-P}}$ [30]. At P = 1, there is an ideal capacitance corresponding to a smooth surface, and at 0.75 < P < 1, there is a real rough surface. The high-frequency resistance R_s_ simulates the electrolyte resistance. The presence of the film adds a Z_pore_ element to the equivalent circuit that simulates the passage of electric current through a porous non-conductive medium filled with electrolyte to the electrochemically active surface of the polarizable electrode. The impedance of this element around medium and high frequencies (f = 10^1^–10^4^ Hz) is equivalent to the impedance of the transmission line [31], which can also be simulated by the CPE element, the degree index of which is P = 0.5. Thus, an equivalent circuit (Figure 9b), which allows for the ability to adequately model and describe the studied systems, was proposed. The spectra for all samples were taken at OCP equal to 0.25 ± 0.05 V.

Figure 9a shows that pure tungsten had the lowest value of polarization resistance R_p_ of the order of 3 × 10^4^ Ohm (low-frequency limit Z at f→0), whereas both types of films had approximately the same polarization resistance R_p_ ≈ 5 × 10^5^ Ohm—almost an order and a half higher. However, the quality of their films differed, which was reflected in the impedance spectra in the region of medium and high frequencies (f = 10^1^–10^4^ Hz). From the Bode diagram, it is clear that the spectrum of 5 min of CPD lay higher than for the prolonged exposure in INFOR.

Table 2 shows that the parameters of the elements of the equivalent circuit for pure W and inhibited polymer films occurred on the tungsten for both of the film formation techniques.

These data show that the value of the parameter Z-T_pore for the sample of 5 min of CPD was by two orders of magnitude smaller than that for the electrode of 60 min of exposure, and the complex resistance inversely proportional to Z-T_pore for the sample of 5 min of CPD was by two orders of magnitude larger, which indicated a lower porosity and, consequently, a better quality of coating. The values of the other parameters had no statistically significant differences.

#### 2.6.2. Corrosion Resistance of Films

The corrosion behavior of tungsten samples depending on the method of film formation are shown in Figure 10.

Figure 10 shows that after 60 days of exposure of pure tungsten in the chamber with periodic condensation of heat and moisture on its surface, uniform corrosion with isolated local damages was observed (Clean W). The time before the first corrosion damage appearance was 4 days. The percentage of damaged surface was 3%. The film that was formed on the tungsten surface by soaking the sample in the aqueous solution of MS + BTA + HEDP for prolonged exposure of 60 min initiated a slowing of corrosion processes (60 min of exposure). In this case, the affected surface area was 1.5%. The spots of corrosion products on the tungsten samples appeared after 18 days of exposure in a chamber with periodic condensation of heat and moisture and did not change in size during the 60 day test. The film, which was formed on tungsten with the CPD method, had a comparably better protective ability compared to the prolonged exposure (5 min of CPD). The corrosion damage proportion was less than 0.5%. The first defects appeared after 25 days of tests. 

## 3. Discussion

As previously stated, there is no information in the literature about the effective inhibitors of tungsten corrosion. The material presented in this article expands the area of application of inhibited formulations (INFOR), particularly of the organosilane class. It was shown that from the aqueous solution of INFOR, there was a receivable protective inhibited polymer film by cataphoresis deposition on the tungsten surface. A negative electrical charge (cathode) was applied to the tungsten product, and a positive charge was applied to the anode. Oxygen was released at the anode, and hydrogen was actively released at the cathode. Under the action of an electric current, the molecules of the inhibited formulations migrated to the surface of the tungsten and were evenly distributed on it. Thus, the formation of the inhibited polymer film from the INFOR aqueous solution proceeded.

Methacryloyloxypropyltrimethoxysilane was proposed as an organosilane. This organosilane has a long chain, and also in its structure, there is a methacrylic group, which makes it a more effective surface modifier and crosslinking agent in comparison with other organosilanes (methoxysilane, vinyltrimethoxysilane) [13,14,15]. The adsorption type corrosion inhibitors proposed to be used as corrosion inhibitors are as follows: 1,2,3-benzotriazole (BTA) and hydroxyethylidene diphosphonic acid (HEDP). Heterocyclic compounds are widely used to protect metals from corrosion damage, because of their ability to chemisorbed on metal surfaces and to form insoluble nanosized protective films. BTA is one of the cheapest, most available, and most researched heterocyclic inhibitor that is widely used for the protection of metals. Considering this, the molecule of BTA is a weak NH-acid that is capable of the hydrolytic polycondensation reaction with silanol groups by reaction 3 (Figure 11) [32,33]. The interaction results in the formation of siloxane–azole fragments that provide additional cross-linking of surface-adsorbed molecules [13,15]. Multibasic phosphonic acids are acidic catalysts of the hydrolytic condensation reaction in the organosilane polymerization. HEDP is a polybasic acid that is capable of creating polymer films on metals when interacting with organosilanes, and is also an effective inhibitor of metal corrosion in aqueous solutions [15,34]. In addition, the presence of additives in the aqueous organosilane solution by way of corrosion inhibitors increases the electrical conductivity of the solution and makes it possible to realize cataphoretic deposition [26,27,28,35].

Using cataphoresis deposition, the process of forming the inhibited polymer films on the tungsten surface from the developed aqueous solution has been optimized as compared to the immersion/exposure method. The process is 12 times faster. Using scanning electron microscopy, we were able to determine the optimal modes of obtaining a continuous polymer film when using cataphoresis deposition. The duration of the CPD is 5 min, and the value of the current density i_min_ = 0.5 A/dm^2^.

Applying the ATR-FTIR technique showed that both cataphoresis deposition and sample soaking in a modifying aqueous solution resulted in the hydrolysis of organosilanes with their subsequent polycondensation, which allows for the assumption that the film formation mechanism under CPD does not change as compared to the prolonged exposure of the tungsten specimens in aqueous solution containing organosilane and corrosion inhibitors (Figure 11) [13,15,35].

Using differential scanning calorimetry, transmission electron microscopy, measurements of the energetic characteristics of the formed films, and adhesion tests, it was possible to study the structure formation and influence of the permolecular structure of films obtained by two methods on the energy characteristics and adhesion properties in the investigated systems. From the data obtained, it was established that the method of the prolonged exposure of tungsten in the INFOR aqueous solution created a denser chemical grid due to the emergence of BTA crystals on the surface. As for CPD, BTA crystals are in the mass of film, which disrupts the crosslinking of the film and reduces its density. Furthermore, the films produced by 60 min of exposure of tungsten specimens in the INFOR solution had higher energetic and adhesion values.

In the investigation of the protective filming ability (EIS and corrosion behavior), the better efficiency of the films produced on the tungsten by CPD was shown. This result is related to the fact that when the sample was exposed to an aqueous solution containing organosilane and corrosion inhibitors, the places where the BTA crystals exited were defective. In this regard, in contact with an aggressive environment, the films obtained through a 60 min exposure showed worse results in comparison with a 5 min CPD.

Thus, in case it is necessary to increase the rate of film formation, the suggested technique of film formation on tungsten as well as the electrolyte developed can be used for the protection of tungsten products from atmospheric corrosion.

## 4. Materials and Methods

### 4.1. Materials

#### 4.1.1. Substrate Materials

In this work, M0 copper plates with a total area of 40 cm^2^ were used, on which tungsten coatings were applied by chemical vapor deposition [36] from a mixture of gases, namely, tungsten hexafluoride and hydrogen at 550 °C. The tungsten coatings had the following properties: the average coating thickness reached 120 µm, porosity did not exceed 0.04%, and the average surface roughness was 6 µm. 

Electrochemical impedance measurements were carried out on disk electrodes. For this purpose, cylindrical copper samples were made, on which tungsten coatings were similarly applied. The lateral surface of the cylinders was covered with insulating material so that the end face of the cylinder disk remained free; the visible surface area was 1.13 cm^2^.

#### 4.1.2. Composition of the INFOR Aqueous Solution 

The aqueous solution was prepared using the following:−methacryloxypropyltrimetoxysilane—MS (Eksport-import, Moscow, Russia) (Figure 12). This organosilane has a long chain and also has a methacrylic group in its structure. The concentration of MS was 0.1 M.−1,2,3-benzotriazole—BTA (Henan GP Chemicals Co., Ltd., Zhengzhou, China) and hydroxyethylidene diphosphonic acid—HEDP (Prime Chemical Group, Moscow, Russia) (Figure 12). They were used as adsorption-type corrosion inhibitors. BTA and HEDP contents were constant at 0.01 M and 0.02 M, respectively.

### 4.2. Preparation of the Aqueous Solution

The completeness of the hydrolysis reaction of organosilanes is determined by the pH value of the aqueous solution. According to [37], full hydrolysis of organosilane molecules occurs at a pH range of 2 to 4. The presence of HEDP in the aqueous solution stabilized the pH to 2. To accelerate the hydrolysis of organosilane, additional treatment is worth using in addition to adjusting the pH of the environment. In this work, ultrasonic treatment (UT) was used. Thus, the INFOR aqueous solution consisting of MS + BTA + HEDP was subjected to UT using Sapphire-0.8 TTs (Sapphire, Moscow, Russia) for 15 min.

### 4.3. Techniques for Forming Films on Tungsten Surfaces

Before the formation of inhibited polymer films on the tungsten surface, the samples were degreased in a mixture of acetone and toluene (1:1) (Prime Chemical Group, Moscow, Russia) for 30 min in a Sapphire-0.8 TTs ultrasonic bath (Sapphire, Moscow, Russia). The formation of inhibited polymer films was performed by two methods: cataphoretic deposition and by exposing the metal in a modifying solution.

#### 4.3.1. Prolonged Exposure of the Metal Sample in the INFOR Aqueous Solution 

The tungsten samples were exposed to the INFOR aqueous solution for 5 min and 1 h. Then, the samples were rinsed in isopropanol (Prime Chemical Group, Moscow, Russia) to remove excess siloxane. After, to cure the formed film, the samples were heat treated with SHS-80-01 MK SPU (SKTU SPU, Smolensk, Russia) at 120 °C for 30 min.

#### 4.3.2. Cataphoretic Deposition (CPD)

Using CPD, films were formed on the tungsten surface from the INFOR aqueous solution prepared according to step 2.2. Figure 13 shows the scheme of the cell for CPD [34,35].

A UNIV-20A/120V current source (Impgold, Saint Petersburg, Russia) with the function of stabilizing, maintaining, and adjusting the output current and voltage was used for CPD. The films were deposited at constant current densities of 0.5, 1.0, 1.5, and 2.0 A/dm^2^. The durations of cataphoresis deposition were 1, 2.5, 5, 10, and 20 min. After CPD, the tungsten products were rinsed in alcohol to remove excess siloxane. Then, the metal sample was placed in a SHS-80-01 MK SPU laboratory drying oven (SKTU SPU, Smolensk, Russia) for film condensation under the conditions described in Section 4.3.1.

### 4.4. Scanning Electron Microscopy (SEM) 

The film surface morphology and coating thickness evaluation were carried out using SEM and EDX. A PSEM-500 (Philips, Amsterdam, The Netherlands) equipped with Wineds energy-dispersive X-ray attachment (Eumex, Heidenrod, Germany) was used in the work. The experimental process was secondary electrons, accelerating voltage of 15 keV, and a spectrum accumulation time for EDX 120 s.

### 4.5. Attenuated Total Reflectance Fourier Transform Infrared Spectroscopy (ATR-FTIR)

The molecular behavior of the ingredients of the inhibited polymer films made by different methods was analyzed using ATR-FTIR spectroscopy. For this aim, we used a Nicolet iN10 infrared microscope (Thermo Fisher Scientific, Waltham, MA, USA) using the ATR mode with Ge-crystal, a research range of 675 to 4000 cm^−1^, the transmission mode (a resolution of 4 cm^−1^ by accumulating 128 scans), and a subsequent processing of spectra in the software Omnic 9 (Thermo Fisher Scientific, Waltham, MA, USA).

### 4.6. Transmission Electron Microscopy (TEM)

The subprimary structure of tungsten films was investigated by the TEM research. For this purpose, we applied the transmission electron microscope EM-301 (Philips, Amsterdam, The Netherlands), the replica method, and etching samples of 20 min in oxygen discharge plasma [38].

### 4.7. Differential Scanning Calorimetry (DSC)

The change in temperature of the phase transitions of the original MS and the inhibited polymer films organized on the tungsten surface by two methods were explored. For these tests, we used the differential scanning calorimeter Netzsch DSC 204F1 (Netzsch, Selb, Germany), a temperature range of –80 to 120 °C, and a heating rate of 10 °C × min^−1^; the first-order derivative of the heat flow was used in further calculations (on account with a small amount of solution).

### 4.8. Surface Free Energy Definition

The energy characteristics of the films were measured using the sessile drop method. Wetting angle measurements were performed at a tensiometer EasyDrop Standard (Kruss, Hamburg, Germany). The following mix of standard test liquids was used: water, glycerine, formamide, dimethylsulfoxide, o-tricresilphosphate; T = 25 (±2) °C. This technique is described in more detail in our previous work [39].

### 4.9. Adhesion Tests

The adhesion properties of the films were investigated using a Zwick/Roell-Z010 series Allround-Line tear machine (Zwick/Roell Gmb and Co., Ulm, Germany) equipped with a 2.5 kN force transducer (Zwick/Roell Gmb and Co., Ulm, Germany), accuracy class 0.5. Studies were performed by comparative analysis of the results during the peeling of the test tape (Dielectric polymers, Inc., Chicago, IL, USA) with silicon-containing adhesive at an angle of 180 degrees at a speed of 100 mm/min from the films surfaces obtained by the two methods.

### 4.10. Protective Properties of Films

#### 4.10.1. Electrochemical Investigations

Exploring of the corrosion-electrochemical behavior of the pure tungsten and films generated on the metal depending on the method of their production were made in a three-electrode cell (the pure tungsten and tungsten with films—work electrode; the platinum wire—counter electrode; reference electrode—silver chloride electrode). The 3.5% NaCl was used as the background solution. A Luggin capillary was brought to the end surface of the work electrode. Electrochemical measurements were carried out on a P-45X potentiostat equipped with an impedance module FRA-24M (Electrochemical Instruments, Chernogolovka, Russia).

First, open-circuit potential (OCP) was measured for 1800 s, and then the electrode was potentiostatized for 300 s at a potential equal to the last measured value of OCP. After bringing the electrode to a steady state, the electrochemical impedance spectrum was measured, also in potentiostatic mode, at the same OCP potential, in the range 10^5^–10^−2^ Hz, amplitude 10 mV.

Four samples of each type were used for each type of test: tungsten without film (pure), prolonged exposure (60 min) in a modifying solution, and 5 min of CPD.

The protective effect of inhibited polymer films was calculated according to Equation (1):(1)$$Z=\frac{R_{p}^{film}-R_{p}^{W}}{R_{p}^{film}}\times 100\%$$

#### 4.10.2. Corrosion Tests

Corrosion tests of tungsten specimens with polymer-inhibited films depending on the method of their production were carried out in a chamber with periodic condensation of moisture from 3.5% NaCl solution [40]. Tungsten-coated plates described in Section 4.1.1 were used as the test material. We maintained 100% relative air humidity in the chamber. The tests were realized at 25 (±2) °C. The time of testing was 60 days. After testing, we evaluated the corrosion damage proportion in compliance with the ASTM D 610-08 [41].

## 5. Conclusions

Using scanning electron microscopy, the optimal modes of obtaining continuous and defect-free inhibited polymer films on the tungsten surface by cataphoresis deposition were determined: the duration of the CPD was 5 min, and the value of current density applied to the sample was 0.5 A/dm^2^.By means of transmission electron microscopy and differential scanning calorimetry, it was established that in the case of 60 min sample soaking in modifying solution, crystal structures of BTA came to the surface that led to the creation of a denser chemical bond grid.The adhesion tests showed that both in the case of 60 min exposure of a sample in the INFOR solution and for the CPD method, the adhesion strength of the film to the metal substrate was not lower than 330 N/m.The use of impedance spectroscopy and corrosion studies demonstrated a better efficiency of films formed on tungsten by CPD.

## 6. Patents

The results of this work resulted in a patent: Gladkikh, N.A.; Dushik, V.V.; Shaporenkov, A.A.; Shapagin, A.V.; Makarychev, Yu.B.; Gordeev, A.V.; Marshakov, A.I. Water suspension containing organosilan, corrosion inhibitor and polycondensation promoter and method for producing protective films on surface of tungsten and coatings on its basis from water suspension containing organosilan, corrosion inhibitor and polycondensation promoter. Patent RU2744336C1, 2021, 14 p.

## Figures and Tables

**Figure 1 ijms-24-14412-f001:**
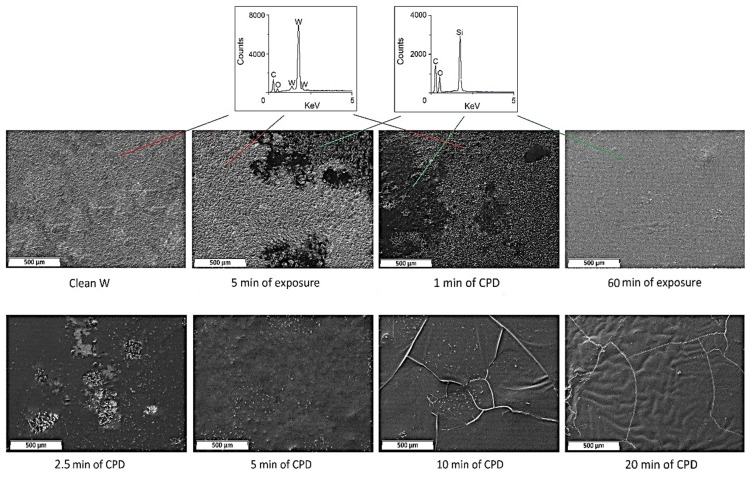
SEM images of films according to the duration of the CPD.

**Figure 2 ijms-24-14412-f002:**
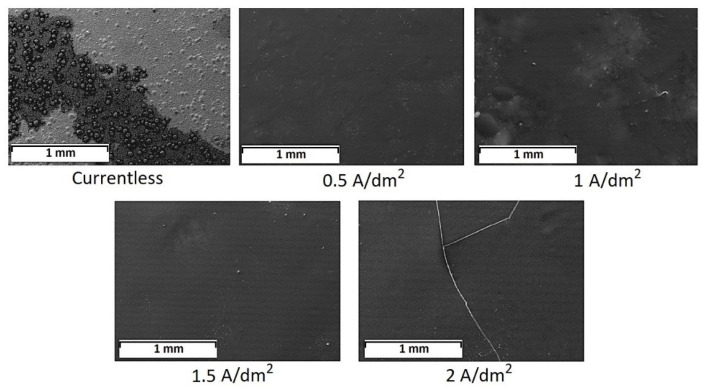
SEM images of films formed as a function of the current density. The duration of the CPD was 5 min.

**Figure 3 ijms-24-14412-f003:**
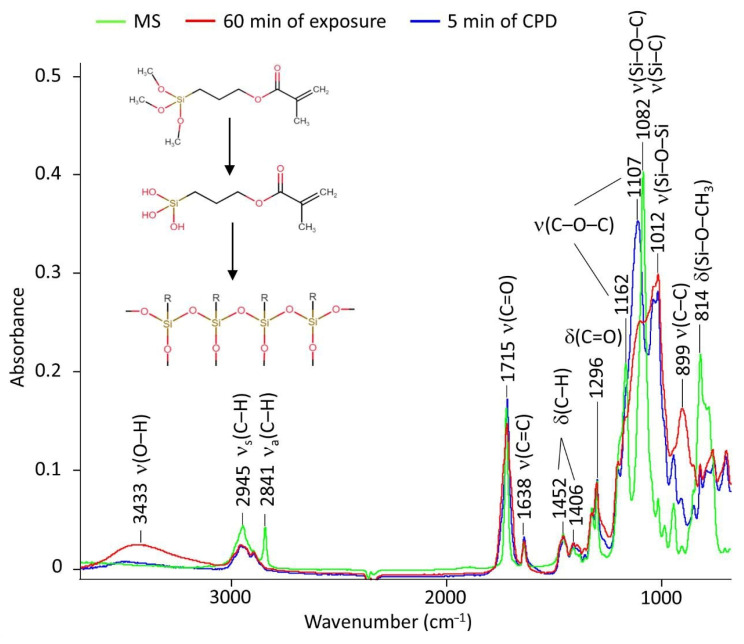
The FTIR data of pure organosilane (green) and the spectrums of inhibited polymer films occurred on tungsten with 60 min of exposure (red) or 5 min of CPD (blue) in the range from 675 to 4000 cm^−1^.

**Figure 4 ijms-24-14412-f004:**
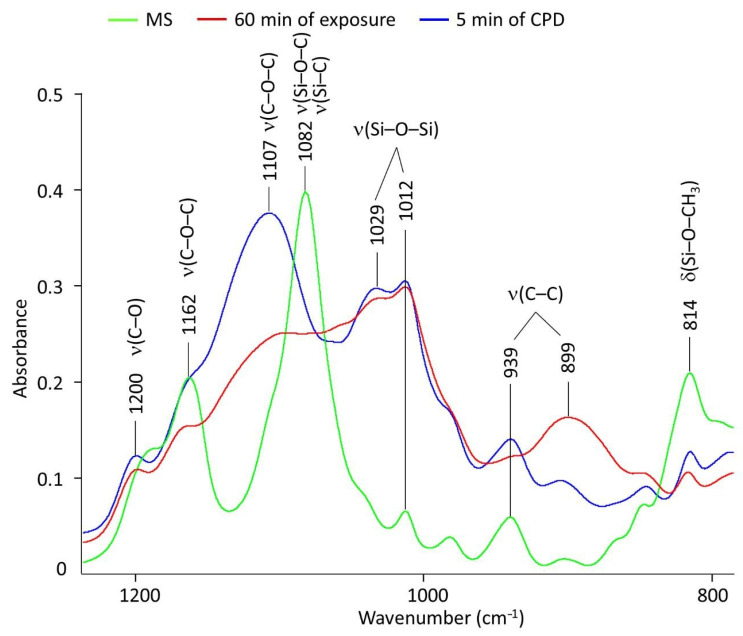
FTIR spectrum of pure MS and spectrums of the inhibited polymer films in area from 800 to 1200 cm^−1^.

**Figure 5 ijms-24-14412-f005:**
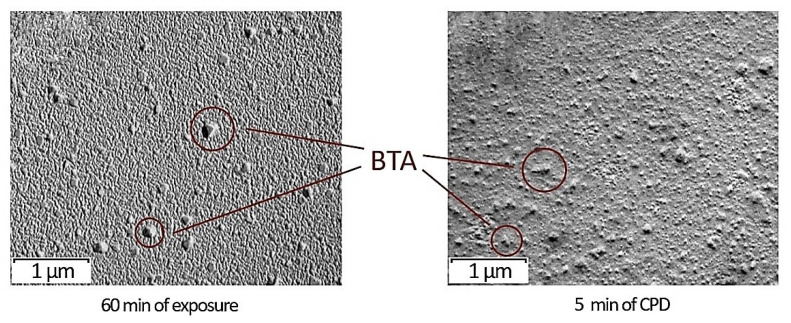
TEM images of inhibited polymer films occurring on the tungsten surface, depending on the film formation method.

**Figure 6 ijms-24-14412-f006:**
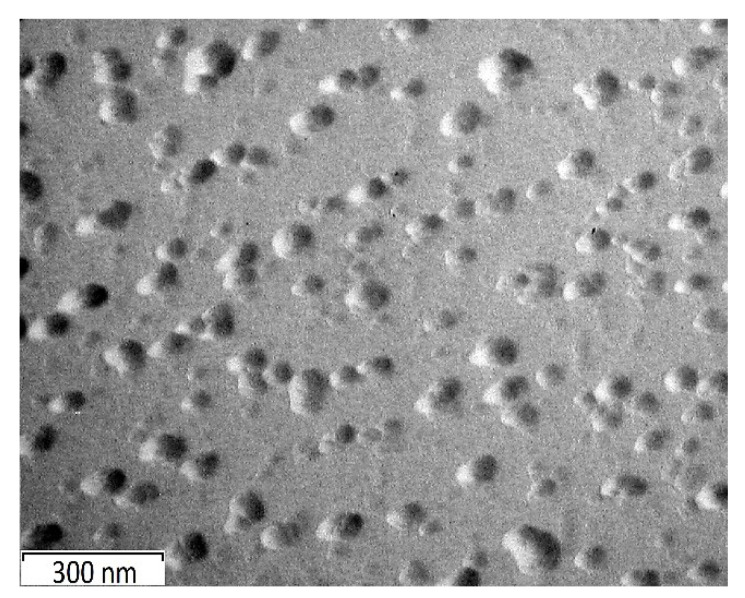
TEM image of BTA crystals from aqueous solutions.

**Figure 7 ijms-24-14412-f007:**
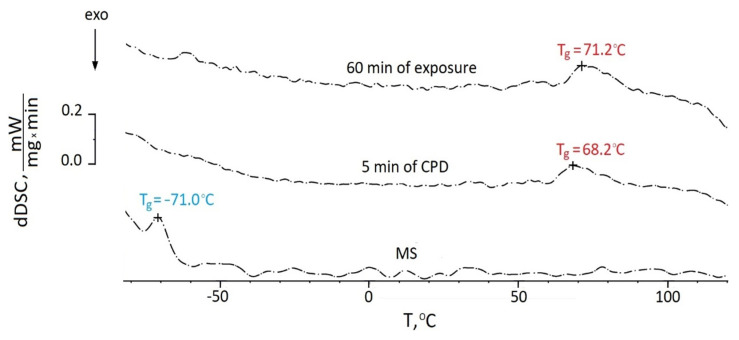
DSC thermograms of the films produced by different methods.

**Figure 8 ijms-24-14412-f008:**
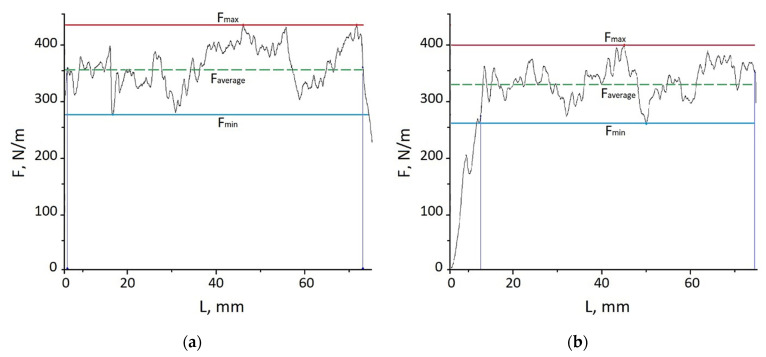
Adhesion characteristics of films depending on the method of film formation: (**a**) 60 min of exposure; (**b**) 5 min of CPD.

**Figure 9 ijms-24-14412-f009:**
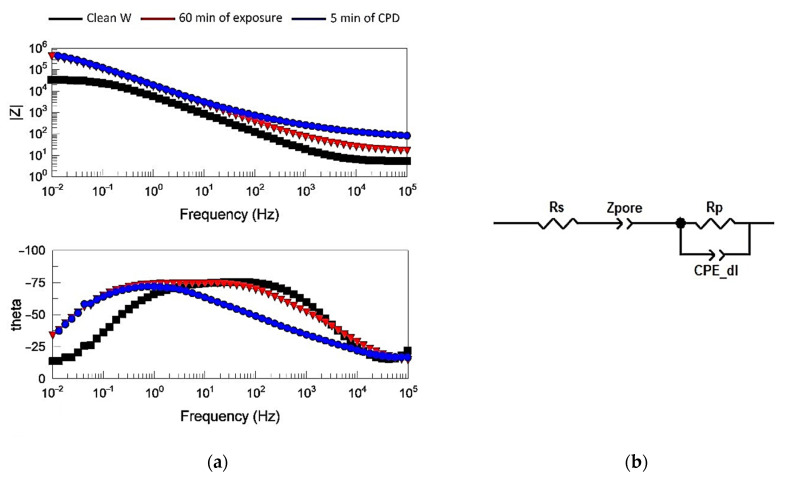
Results of the films’ electrochemical behavior study: (**a**) typical Bode diagrams, spectrum of pure tungsten, and films formed on the electrodes using different methods; (**b**) equivalent circuit.

**Figure 10 ijms-24-14412-f010:**
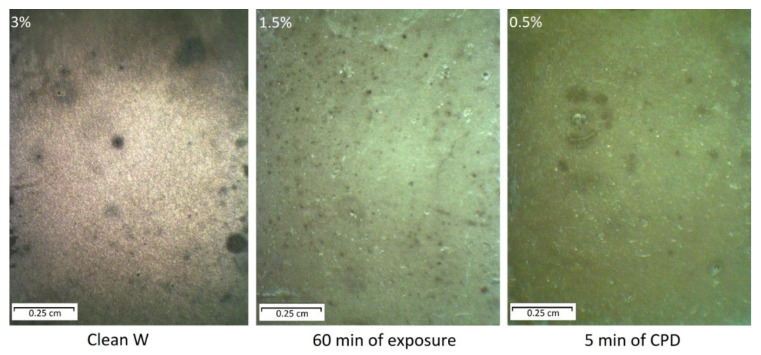
Appearance of tungsten specimens after their testing for 60 days, also indicating the corrosion damages proportion.

**Figure 11 ijms-24-14412-f011:**
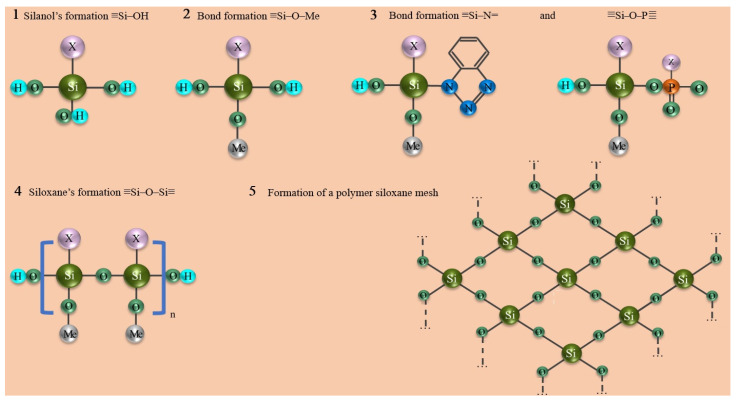
Film formation reactions.

**Figure 12 ijms-24-14412-f012:**
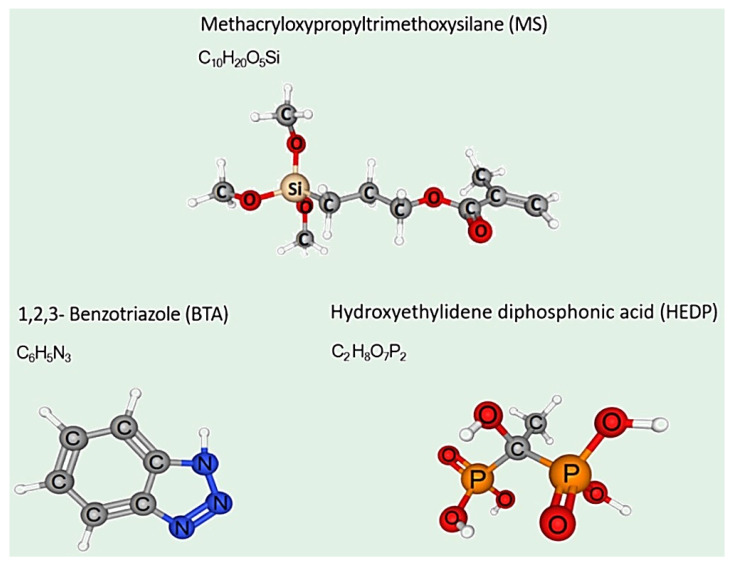
The structural formulas of the substances.

**Figure 13 ijms-24-14412-f013:**
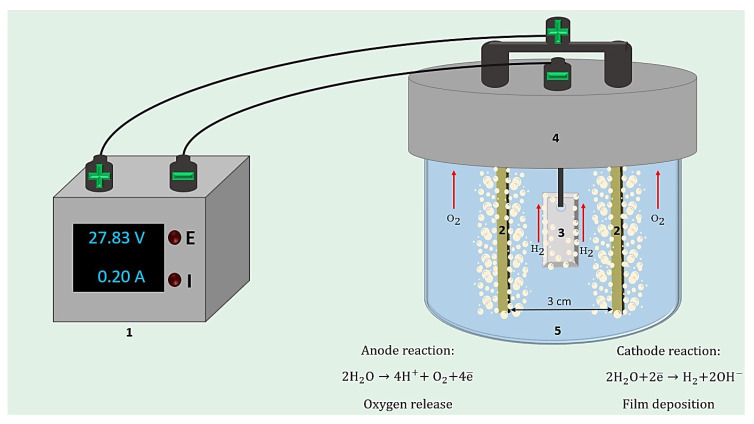
The scheme of the cell for CPD: 1—supply source; 2—the stainless anodes; 3—the tungsten product; 4—electrode holder; 5—the INFOR aqueous solution.

**Table 1 ijms-24-14412-t001:** Surface energy of films obtained by exposure of the sample into modifying solution for 60 min and by CPD for 5 min.

Sample	γ_O_, mJ/m^2^	γ_P_, mJ/m^2^	γ_D_, mJ/m^2^
60 min of exposure	40.23	4.70	37.53
5 min of CPD	37.20	5.11	32.09
Difference Δ	5.03	0.41	5.44

**Table 2 ijms-24-14412-t002:** The averaged values of the calculation of the parameters of the equivalent circuit.

Materials	Rs, Ohm	Z-T_pore, s^P^/Ohm	Z-P_pore	Rp, Ohm	CPE-T_dl, s^P^/Ohm	CPE-P_dl	Z, %
Clean W	6.9 ± 1.1	–	–	(3.1 ± 1.1) × 10^4^	(4.3 ± 1.5) × 10^−5^	0.84 ± 0.02	–
60 min of exposure	10.2 ± 2.8	(2.1 ± 0.7) × 10^−3^	0.42 ± 0.06	(3.6 ± 1.4) × 10^5^	(1.6 ± 0.2) × 10^−5^	0.92 ± 0.05	91.41
5 min of CPD	12.8 ± 3.2	(4.9 ± 1.8) × 10^−5^	0.47 ± 0.09	(6.1 ± 2.8) × 10^5^	(1.4 ± 0.6) × 10^−5^	0.95 ± 0.05	96.73

## Data Availability

Not applicable.

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
