# Peer review of "Methods of Formation of Protective Inhibited Polymer Films on Tungsten"

_ijms, 2023, doi:10.3390/ijms241914412_

Round 1
Reviewer 1 Report
The authors have presented a high quality manuscript which I admire and suggest to be accepted with very minor revision.
One minor comment is to move (even though extensive) Materials and Methods section prior to results.
Also the definition of size needs rectification, as it is shown for instance like solid film with a thickness of 19 ÷ 24 µm, which should be written as 19 - 24 µm instead. This must be corrected throughout the manuscript.
Good luck!
The quality of English is fine and does not require additional proofreading.
Reviewer 2 Report
The authors did a very nice work and got some good results. I suggest it accept after minor revision.
1. The unit of CPE should be checked.
Good
Reviewer 3 Report
The paper describes a new cataphoretic application of Silanoles on Tungsten for corrosion protection. Basically a new and interesting approach.
Some issues has to be improved as follows:
Discussion to fig. 3:
In the discussion changes of intensities are discussed without a normalization of the spectra. Because of the fsact that ATR spectra shows no correlation to the intensity the spectra have to be normalized to discuss intensity changes.
Discussion to fig. 4:
It is pointed out, that the 10% amount of HEDP is not visible in the spectra but several signals in the range of 900-1100 cm-1 are visible and could be caused by P-OH groups. An experiment without HEDP in the film proofes it and has to be performed.
Fig.11:
BTC should react with the free Silanol but Silanoles does not react with phosphonates. Why?
The kataphoretic deposition process is not explained. Why precipitate the polysiloxane on the surface with the additives? If the HEDP is not visible, it may not deposit on the surface.
